# Distinct inflammatory and wound healing responses to complex caudal fin injuries of larval zebrafish

Veronika Miskolci[1†], Jayne Squirrell[2†], Julie Rindy[1], William Vincent[1,3], John Demian Sauer[1], Angela Gibson[4], Kevin W Eliceiri[2], Anna Huttenlocher[1,5*]

[1]Department of Medical Microbiology and Immunology, University of Wisconsin-Madison, Madison, United States; [2]Laboratory for Optical and Computational Instrumentation, University of Wisconsin-Madison, Madison, United States; [3]First Year Experience, Northern Arizona University, Flagstaff, United States; [4]Department of Surgery, University of Wisconsin-Madison, Madison, United States; [5]Department of Pediatrics, University of Wisconsin-Madison, Madison, United States

*For correspondence:
huttenlocher@wisc.edu

†These authors contributed equally to this work

Competing interests: The authors declare that no competing interests exist.

**Abstract** Wound repair is controlled temporally and spatially to restore tissue homeostasis. Previously we reported that thermal damage of the larval zebrafish fin disrupts collagen organization and wound healing compared to tail transection (LeBert et al., 2018). Here we characterize different injury models in larval zebrafish to dissect temporal and spatial dynamics of repair in complex damage. We found that each damage model triggers distinct inflammatory and tissue responses, with Stat3 and TGFβ playing key roles in the regulation of mesenchymal cells during simple repair. While thermal injury disrupts collagen fibers initially, healing is recovered as inflammation resolves, and mesenchymal cells and collagen fibers align. By contrast, infected wounds lead to persistent inflammation and loss of mesenchymal cells, resulting in minimal tissue repair. These wound models have broad physiological relevance, thereby providing a valuable advance in our toolkit to probe the dynamics of inflammation and wound repair in complex tissue damage.
DOI: https://doi.org/10.7554/eLife.45976.001

## Introduction

Wound repair requires the integration of complex cellular networks and extracellular matrix remodeling to mediate resolution of damage. Caudal fin transection or sterile needle wounding of larval zebrafish have become invaluable in vivo models of inflammation and wound repair (*LeBert and Huttenlocher, 2014*; *Renshaw et al., 2006*). These models have led to key insights into early wound signals (*Niethammer et al., 2009*; *Yoo et al., 2012*; *Yoo et al., 2011*) and mechanisms of resolution of inflammation (*Mathias et al., 2006*; *Tauzin et al., 2014*) due, in part, to the unique capability for monitoring the wound response over a long duration. These sterile wounds are akin to controlled surgical wounds, lacking complexities associated with traumatic injuries or complications such as infections (*Velnar et al., 2009*). We recently introduced a burn wound assay that disrupts collagen organization to study the role of collagen-based epithelial projections during wound repair (*LeBert et al., 2018*). We found that, unlike the simple transection, thermal injury at the caudal fin led to a dramatic loss of collagen fibers in the wound region, accompanied by a marked impact on healing progression at 48 hr post wound (hpw). This unexpected finding provided an opportunity for exploring alternate caudal fin injuries with the goal of developing wound models with added clinical and physiological relevance. Here, we characterize the kinetics of repair in the thermal injury model in comparison to the simple transection. Additionally, we examined how the presence of infection

alters healing dynamics following a simple transection by applying *L. monocytogenes* bacteria in the caudal fin transection model. We evaluated three aspects of these injuries (*Figure 1A*), representing key stages of wound healing – new tissue formation, inflammation and remodeling (*Ellis et al., 2018*). We also began to investigate pathways that regulate the dynamics of tissue reorganization, and identified a role for Stat3 and TGFβ. Thermal injury and infected transection elicited different responses compared to the simple transection in all aspects that we characterized, allowing for new insights into the mechanisms of wound repair.

## Results and discussion

### Thermal injury delays, while infection impairs wound repair

To begin to characterize these new caudal fin wound models, we first assessed the healing capacity by measuring tissue regrowth area of the caudal fin following injury. We performed caudal fin wounding on wild-type larvae at 3 days post fertilization (dpf) and monitored individual larvae daily by bright-field imaging. The healing dynamics following burn wound or *L. monocytogenes* (*Lm*)-infected transection were different from simple transection (*Figure 1*). Wound healing in the burn wound was attenuated between 24 and 72 hpw, but unexpectedly recovered such that the tissue regrowth area was comparable to that of simple transection by 96 hpw (*Figure 1B,D*). This observation suggested that events leading up to 72 hpw were critical to facilitating the healing process. In contrast, we did not observe similar recovery following transection in the presence of *L. monocytogenes*. The tissue regrowth area post *Lm*-infected transection remained markedly diminished relative to simple transection, even at 168 hpw (7 days post infection, dpi) (*Figure 1C,E*). Importantly, the infection had no measurable impact on the survival of larvae (*Figure 1—figure supplement 1D*). We monitored the course of infection after rinsing away bacteria. We found that bacteria entered at the wound site and expanded, peaking at 72–96 hpw, and were largely cleared from the host by 120–168 hpw (*Figure 1—figure supplement 1A*). Although the infection remained localized, limited dissemination of bacteria occurred in small patches along the neural tube (17 out of 47 larvae, three biological replicates) (*Figure 1—figure supplement 1C*). *L. monocytogenes* did not infect unwounded larvae (*Figure 1—figure supplement 1B*). Overall, while the burn wound was ultimately able to recover and heal, the damage caused by *L. monocytogenes* infection was sufficient to lead to a significant and long-lasting defect in wound healing.

### Thermal injury and infection trigger distinct inflammatory responses

The differential capacity for repair suggested that the inflammation in response to the burn wound or *Lm*-infected transection may also vary from the simple transection. To characterize the inflammatory response, we quantified neutrophil and macrophage recruitment to the wound site. Although the inflammation in the burn wound followed a similar trend as that in the simple transection, the burn wound recruited significantly more neutrophils and macrophages (*Figure 2A–C*; *Figure 2—figure supplement 1*). Interestingly, while neutrophil recruitment peaked at different times, it appeared to resolve by 24 hpw in both conditions (*Figure 2B*; *Figure 2—figure supplement 2A*). The number of macrophages reached peak levels at 6 hpw and were clearing by 96 hpw in both injuries (*Figure 2C*; *Figure 2—figure supplement 2B*). The leukocyte infiltration at the wound site over time appeared to be related to wound repair, as we detected minimal recruitment at the caudal fin of age-matched unwounded larvae (*Figure 2—figure supplement 2C,D*). Furthermore, we monitored differentially activated subpopulation of macrophages using a transgenic reporter line for TNFα expression (*Marjoram et al., 2015*) to identify pro-inflammatory M1-like cells (*Kratochvill et al., 2015*; *Murray, 2017*). We considered TNFα-positive macrophages to be M1-like macrophages as previously reported (*Nguyen-Chi et al., 2015*), and TNFα-negative as a differentially activated macrophage population. This TNFα-negative population may represent the anti-inflammatory M2-like cells (*Murray, 2017*) however, due to the current lack of a reporter line for M2-like macrophages in zebrafish, we simply distinguished macrophage populations as TNFα-positive or negative. The response appeared to be more pro-inflammatory in the burn wound (*Figure 2A,C*). While mostly TNFα-negative macrophages are recruited to a simple transection throughout the course of the

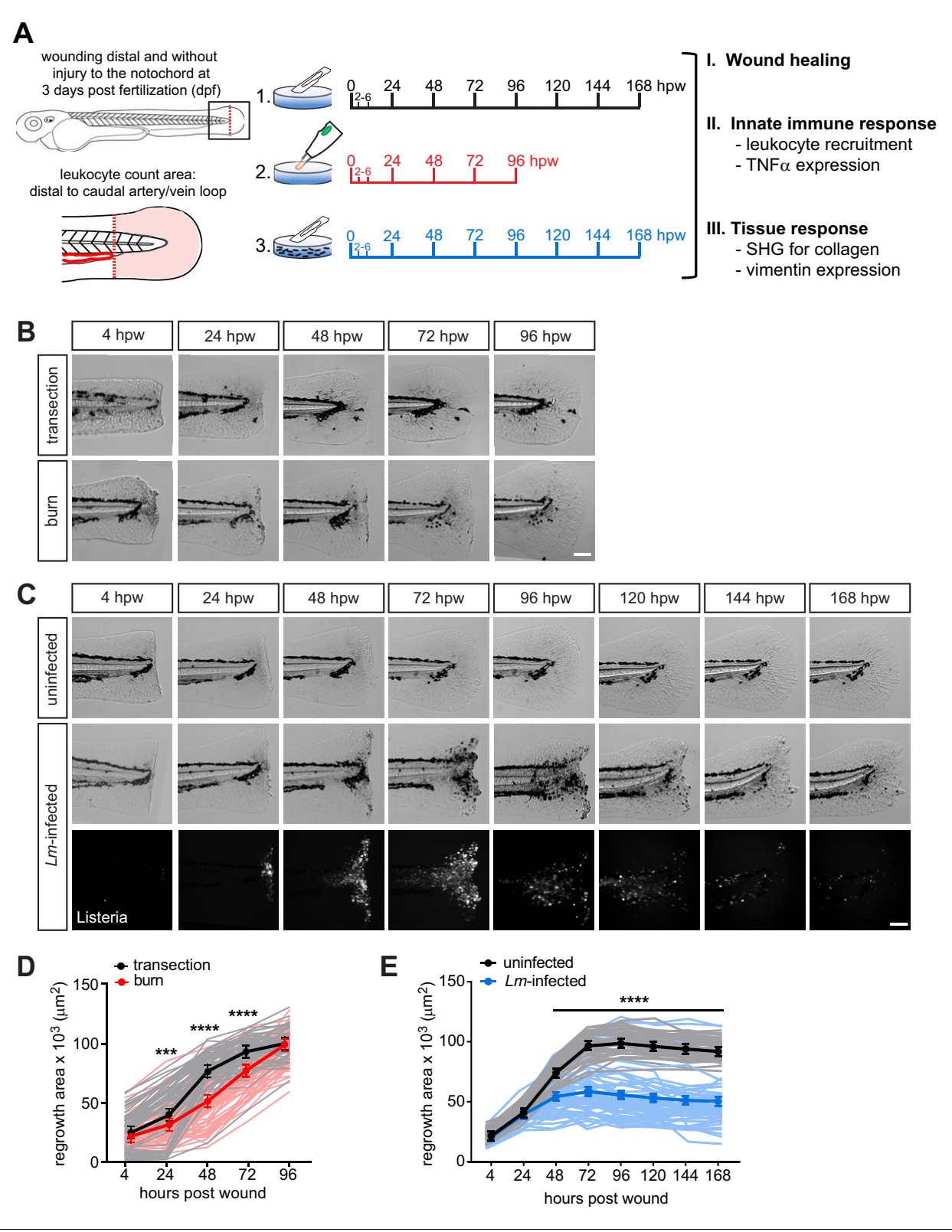

**Figure 1.** Caudal fin recovers from thermal injury, while wound healing is impaired in the presence of infection. (A) Experimental schematic and analyses. (B) Single-plane brightfield images of caudal fin area of individual wild-type larvae over time in response to simple transection or thermal injury, and in (D) the corresponding quantification of tissue regrowth area. Values are least square means and SE from three biological replicates with associated p values. Total N = 62–71 larvae per time point for each treatment. (C) Single-plane brightfield or fluorescent images of caudal fin area of

*Figure 1 continued on next page*

*Figure 1 continued*

individual wild-type larvae over time in response to uninfected transection or *L. monocytogenes* (*Lm*)-infected transection using mCherry-expressing *L. monocytogenes*, and in (**E**) the corresponding quantification of tissue regrowth area over time are shown. Values are arithmetic means and SE from three biological replicates, with associated p values obtained by analyzing ranks due to residuals not being normally distributed. Total N = 39–58 larvae per time point for each treatment. \*\*\*p<0.001, \*\*\*\*p<0.0001. Lines in lighter color depict values for every larva measured over three biological replicates. Scale bar is 100 microns.

DOI: https://doi.org/10.7554/eLife.45976.002

The following source data, source code and figure supplements are available for figure 1:

**Source data 1.** Related to *Figure 1D*.
DOI: https://doi.org/10.7554/eLife.45976.004
**Source data 2.** Related to *Figure 1E*.
DOI: https://doi.org/10.7554/eLife.45976.005
**Source code 1.** Related to *Figure 1D and E*.
DOI: https://doi.org/10.7554/eLife.45976.006
**Figure supplement 1.** *L. monocytogenes* infection during tail wound infection is associated with minimal dissemination and negligible effect on host survival.
DOI: https://doi.org/10.7554/eLife.45976.003

wound response, the burn wound recruits a more polarized population of macrophages (*Figure 2A, C*), where at least 50% of macrophages in the burn wound were TNFα-positive by 24 hpw. There is also a shift in the proportion of differentially activated macrophages in the burn wound; TNFα-positive macrophages were most prominent 6–48 hpw and began to subside by 72 hpw, while the total number of macrophages was maintained over this time period (*Figure 2C*), and by 96 hpw most macrophages in the burn wound were TNFα-negative. Interestingly, this shift in macrophage activation (72–96 hpw) appears to coincide with the time period when healing begins to recover in the burn wound (72–96 hpw; *Figure 1B,D*). This is consistent with evidence that macrophages undergo a phenotypic switch (switch from M1- to M2-like activation) during the different phases of wound repair to support optimal healing (*Krzyszczyk et al., 2018*; *Novak and Koh, 2013*). The course of inflammation was drastically different in response to the *Lm*-infected transection. Both neutrophil and macrophage infiltration increased over time, to the extent that we were not able to count individual leukocytes reliably beyond 48 hpw (*Figure 2D–F*; *Figure 2—figure supplement 3*). Therefore, we quantified TNFα expression in macrophages by measuring the area of fluorescence signal (*Figure 2—figure supplement 2E*). Unlike in the simple transection, the majority of macrophages were TNFα-positive and the proportion remained similar throughout the time course at the infected tail wound (*Figure 2G*; *Figure 2—figure supplement 2F*). The inflammation began to clear by 168 hpw (7 dpi) (*Figure 2D,G*; *Figure 2—figure supplement 2F*), suggesting that regenerate growth may recover beyond 168 hpw. However, to avoid the complication of feeding larvae, we did not monitor wound repair past this time point. In sum, the burn wound triggered significantly more inflammation relative to the simple transection, while the inflammatory response was most extensive both in magnitude and duration in response to *Lm*-infected transection. In addition, inflammation resolution is initiated in all three injuries, albeit with different temporal dynamics. Moreover, the persistent presence of TNFα-positive macrophages appears to negatively correlate with wound repair.

## Tissue organization recovers from thermal injury while infection disrupts remodeling

We next examined how the tissue response varied amongst the three injuries. Our previous work showed that proper collagen fiber organization is essential for functional wound repair in larval zebrafish (*LeBert et al., 2015*) and that induction of vimentin expression in cells at the wound edge plays an important role in collagen (I and II) production and fiber reorganization (*LeBert et al., 2018*). Based on the observed differences in healing capacity of the wound models, we reasoned that collagen organization and vimentin expression at the wound edge would differ. We examined the impact of the various wounds on collagen fiber organization by performing SHG microscopy. As previously reported (*LeBert et al., 2018*), collagen fibers were completely disrupted upon thermal injury but without the loss of the tissue, as measured by the area devoid of fibers (*Figure 3A*). Collagen fibers began to reappear over time and were clearly visible by 72–96 hpw, extending

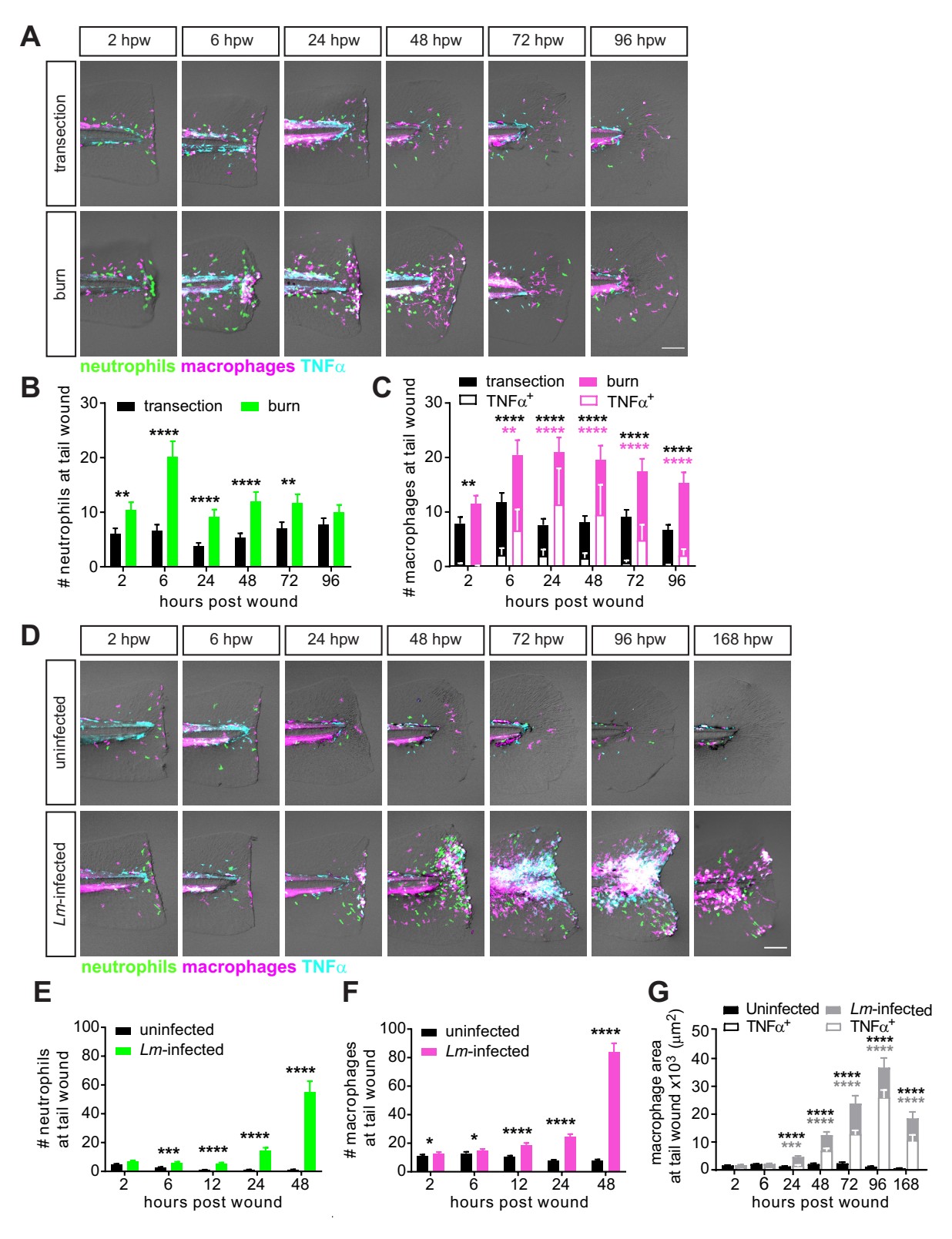

**Figure 2.** Caudal fin injuries trigger distinct inflammatory responses. (**A**) Sum-projections of z-stacks acquired by laser scanning confocal microscope using triple transgenic larvae (*Tg(tnf:GFP) x Tg(lysC:BFP/mpeg1:mCherry-CAAX)*) over time in response to simple transection or thermal injury. Merged channels are displayed; single channels are shown in ***Figure 2—figure supplement 1***. Scale bar is 100 microns. Leukocyte recruitment was quantified by counting (**B**) neutrophils and (**C**) macrophages in the caudal fin tissue distal to the caudal artery/vein loop (***Figure 1A***). In parallel, TNFα expression

*Figure 2 continued on next page*

*Figure 2 continued*

in macrophages was monitored using TNFα reporter and scored as negative or positive for expression, and TNFα-positive counts are shown in (C). Values in (B) and (C) are least square means and SE from four biological replicates, with associated p values. Total N = 28–44 larvae per time point for each treatment. **p<0.01, ****p<0.0001; black and magenta * depict p values for leukocyte and TNFα-positive counts, respectively. p values for comparing time points within each injury are provided in *Figure 2—figure supplement 2A,B*. (D) Sum-projections of z-stacks acquired by laser scanning confocal microscope using triple transgenic larvae (*Tg(tnf:GFP) x Tg(lysC:BFP/mpeg1:mCherry-CAAX)*) over time in response to uninfected or *Lm*-infected transection. Merged channels are displayed; single channels are shown in *Figure 2—figure supplement 3*. Scale bar is 100 microns. Leukocyte recruitment was quantified using double transgenic larvae (*Tg(lysC:mCherry-histone2b) x Tg(mpeg1:GFP-histone2b)*) where (E) neutrophils and (F) macrophages were counted in caudal fin area distal to the caudal artery/vein loop in single-plane images acquired by Zeiss Zoomscope. TNFα expression in macrophages (G) was quantified by area thresholding (see Materials and methods and *Figure 2—figure supplement 2E*) using images in (D). For clarity, values in (G) for the uninfected transection are also shown separately in *Figure 2—figure supplement 2F*. Values in (E) and (F) are least square means and SE from four biological replicates, with associated p values. Values for macrophages were fitted with poisson distribution. Total N = 30–60 larvae per time point for each treatment. *p<0.05, ***p<0.001, ****p<0.0001. Values in (G) are arithmetic means and SE from three experimental replicates with associated p values obtained by analyzing ranks due to residuals not being normally distributed. Total N = 12–27 larvae per time point for each treatment. ***p<0.001, ****p<0.0001; black and gray * depict p values for macrophage and TNFα-positive areas, respectively.
DOI: https://doi.org/10.7554/eLife.45976.007

The following source data, source code and figure supplements are available for figure 2:

**Source data 1.** Related to *Figure 2B*.
DOI: https://doi.org/10.7554/eLife.45976.011
**Source data 2.** Related to *Figure 2C*.
DOI: https://doi.org/10.7554/eLife.45976.012
**Source data 3.** Related to *Figure 2C*.
DOI: https://doi.org/10.7554/eLife.45976.013
**Source data 4.** Related to *Figure 2E*.
DOI: https://doi.org/10.7554/eLife.45976.014
**Source data 5.** Related to *Figure 2F*.
DOI: https://doi.org/10.7554/eLife.45976.015
**Source data 6.** Related to *Figure 2G*.
DOI: https://doi.org/10.7554/eLife.45976.016
**Source data 7.** Related to *Figure 2G*.
DOI: https://doi.org/10.7554/eLife.45976.017
**Source code 1.** Related to *Figure 2B,C and E*.
DOI: https://doi.org/10.7554/eLife.45976.018
**Source code 2.** Related to *Figure 2F*.
DOI: https://doi.org/10.7554/eLife.45976.019
**Source code 3.** Related to *Figure 2G*.
DOI: https://doi.org/10.7554/eLife.45976.020
**Figure supplement 1.** Leukocyte recruitment in response to thermal injury.
DOI: https://doi.org/10.7554/eLife.45976.008
**Figure supplement 2.** Leukocyte recruitment in the caudal fin tissue is minimal in unwounded larvae.
DOI: https://doi.org/10.7554/eLife.45976.009
**Figure supplement 3.** Leukocyte recruitment in response to transection in the presence of infection.
DOI: https://doi.org/10.7554/eLife.45976.010

progressively outward (*Figure 3A,B*). Despite this initial disruption in collagen integrity, the collagen fiber organization relative to the wound edge was comparable to the simple transection during repair (*Figure 3A,C*). Interestingly, this time line for the progression in collagen fiber reappearance coincides, similarly as the shift in macrophage activation, with the time period when healing recovers in the burn wound (72–96 hpw; *Figure 1B,D*). In the case of the *Lm*-infected transection we observed the opposite pattern: limited damage to collagen fibers but disrupted tissue at the wound area early after wounding (*Figure 3A*). However, dysregulated collagen fiber organization became apparent over time as indicated by the increasing area devoid of collagen fibers (*Figure 3A,D*), as well as fibers becoming disarrayed and less perpendicular to the wound edge (*Figure 3A,E*).

Based on our previous observation of the interplay between collagen and vimentin-expressing cells, we utilized the *vimentin* reporter line (*LeBert et al., 2018*) to assess the presence of vimentin-expressing cells in the different wounds. The number of vimGFP-positive cells at the wound edge

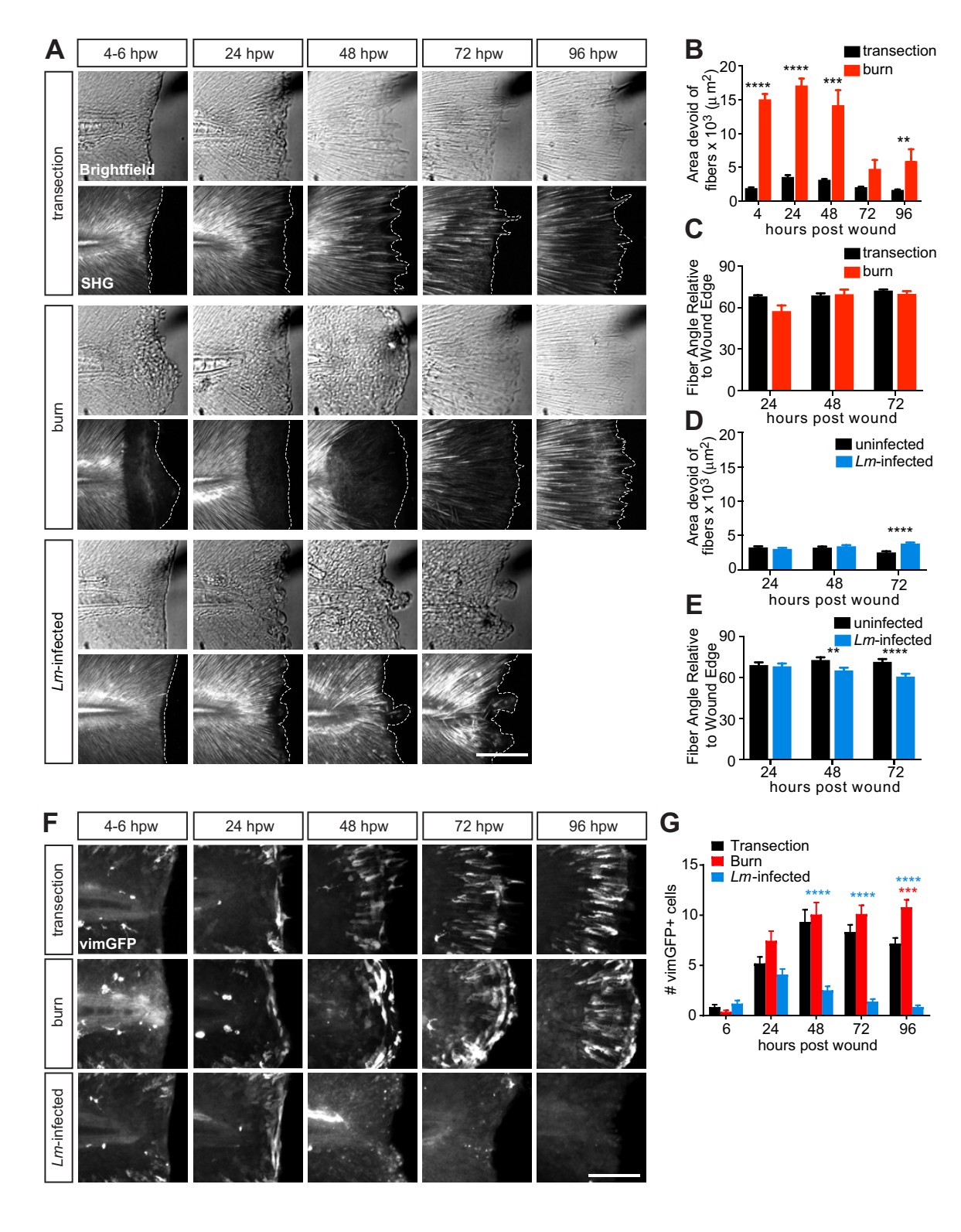

**Figure 3.** Multiphoton microscopy of collagen and vimentin-expressing cells identify differences in tissue remodeling over time amongst the injuries. (A) Z-projections of caudal fins imaged at different times after transection, thermal injury (burn) or infected transection showing the tissue wound edge in the bright field and corresponding SHG image. White dotted outline indicates wound edge from brightfield image. Scale bar is 100 microns. (B) Graph showing the area devoid of SHG fibers (from fiber ends to wound edge) following transection or thermal injury. Values are arithmetic means and

*Figure 3 continued on next page*

*Figure 3 continued*

SE from three biological replicates. p values obtained by analyzing ranks due to residuals not being normally distributed. Total N = 23–28 larvae per time point for each treatment. (**C**) Graph showing the angle of fiber segments relative to the wound edge, as quantified by CurveAlign software (see Materials and methods), within 50 microns of the wound edge with transection or thermal injury. 0° is parallel to the wound edge while 90° is perpendicular to the wound edge. Values are arithmetic means and SE from three experimental replicates. p values obtained by analyzing ranks due to residuals not being normally distributed. Total N = 14–25 larvae per time point for each treatment. (**D**) Graph showing the area devoid of SHG fibers (from fiber ends to wound edge) following uninfected or *Lm*-infected transection. Values are least square means and SE from three biological replicates, with associated p values. Total N = 28-32 larvae per time point for each treatment. (**E**) Graph showing the angle of fiber segments relative to the wound edge in uninfected or *Lm*-infected transections. Values are least square means and SE from three biological replicates, with associated p values. Total N = 17–31 larvae per time point for each treatment. \*\*p<0.01, \*\*\*p<0.001, \*\*\*\*p<0.0001. (**F**) Z-projections of multiphoton microscopy images of vimGFP-positive cells at the wound edge in caudal fins at different times after simple transection, burn or *Lm*-infected transection. Scale bars are 100 microns. Merge of z-projections of vimGFP-positive cell images (green) with z-projections of corresponding SHG images (white) are shown in *Figure 3—figure supplement 1B*. (**G**) Graph showing the number of vimGFP-positive cells within a 50 × 100 micron box adjacent to the wound edge and centered vertically with the notochord (*Figure 4—figure supplement 1A*). Values are least square means and SE from three biological replicates, with associated p values. Total N = 19–28 larvae per time point for each treatment. For clarity, only relevant comparisons (burn in red, or *Lm*-infected transection in blue, to the control transection) are shown on the graph; \*\*\*p<0.001, \*\*\*\*p<0.0001.

DOI: https://doi.org/10.7554/eLife.45976.021

The following source data, source code and figure supplements are available for figure 3:

**Source data 1.** Related to *Figure 3B*.
DOI: https://doi.org/10.7554/eLife.45976.024
**Source data 2.** Related to *Figure 3C*.
DOI: https://doi.org/10.7554/eLife.45976.025
**Source data 3.** Related to *Figure 3D*.
DOI: https://doi.org/10.7554/eLife.45976.026
**Source data 4.** Related to *Figure 3E*.
DOI: https://doi.org/10.7554/eLife.45976.027
**Source data 5.** Related to *Figure 3G*.
DOI: https://doi.org/10.7554/eLife.45976.028
**Source code 1.** Related to *Figure 3B and D*.
DOI: https://doi.org/10.7554/eLife.45976.029
**Source code 2.** Related to *Figure 3B–E*.
DOI: https://doi.org/10.7554/eLife.45976.030
**Source code 3.** Related to *Figure 3G*.
DOI: https://doi.org/10.7554/eLife.45976.031
**Source code 4.** Related to *Figure 3G*.
DOI: https://doi.org/10.7554/eLife.45976.032
**Figure supplement 1.** VimGFP-positive cells at the wound recover over time at the *Lm*-infected tail wound.
DOI: https://doi.org/10.7554/eLife.45976.022
**Figure supplement 2.** Tail wound infection in the presence of *hly* mutant of *L. monocytogenes* causes no delay in vimGFP-positive cells and is associated with limited inflammation.
DOI: https://doi.org/10.7554/eLife.45976.023

was similar between the simple transection and the burn wound, but markedly different in response to the *Lm*-infected transection (*Figure 3F,G*; *Figure 3—figure supplement 1A,B*). The number of vimGFP-positive cells progressively declined to nearly undetectable levels in the *Lm*-infected wound (*Figure 3F,G*). Examination of the *Lm*-infected tail wound at later time points revealed that *vimentin* expression at the wound microenviroment recovers by 120 hpw (*Figure 3—figure supplement 1C, D*). These observations are consistent with, and extend, our previous findings that vimentin supports collagen fiber production and organization during wound repair (*LeBert et al., 2018*). The absence of vimGFP-positive cells at the *Lm*-infected wound temporally correlates with a defect in collagen fiber remodeling accompanied by impaired wound repair. The loss of vimentin-expressing cells at the wound edge in the presence of infection was unexpected. This is likely due to combination of multiple factors, such as cell death and inflammation. Identifying the mechanisms that lead to this effect on vimentin-expressing cells will be the focus of future studies. Interestingly, when we infected with the *hly* mutant of *L. monocytogenes*, a mutant that fails to escape from the phagosome and enter the cytosol of the host cell resulting in attenuated *L. monocytogenes* virulence (*Theisen and*

*Sauer, 2016*), we failed to observe the loss and delay in the appearance of vimGFP-positive cells at the wound edge, while detecting drastically reduced inflammation (*Figure 3—figure supplement 2A,B*) compared to wild-type *L. monocytogenes* infection. These findings indicate that the excessive inflammation associated with *Lm*-infected tail wound may contribute to the loss and delay of the vimentin-expressing cells, providing evidence that there is a crosstalk between the immune response and the presence of vimentin-expressing cells.

## Disrupting Stat3 or TGFβ pathways impairs wound healing and diminishes wound-associated vimentin-expressing cells

Our findings suggest that the appearance of vimentin-expressing cells correlates with repair, suggesting a key role in wound healing. To begin identifying pathways that regulate the dynamics of vimentin-expressing cells at the wound microenvironment and their effects on repair, we examined the role of signal transducer and activator of transcription 3 (STAT3), a transcription factor reported to enhance the expression of the vimentin gene (*Wu et al., 2004*). We tested the role of STAT3 in wound healing following simple transection using the recently developed *stat3* mutant (*Liu et al., 2017b*). Tissue regrowth at 72 hr post tail transection (hptt) of *stat3*$^{stl27/+}$ (stat3+/-) larvae was similar to that of wild-type (stat3+/+) larvae, while it was significantly impaired in *stat3*$^{stl27/stl27}$ (stat3-/-) larvae (*Figure 4A,B*). This defect is most likely wound-specific and not due to hindered development as the *stat3* mutants were reported to exhibit normal development up to 15 dpf (*Liu et al., 2017b*), although we did detect a small reduction in tail fin area of age-matched unwounded larvae (*Figure 4—figure supplement 1A,B*). The inhibition of tissue regrowth following tail transection suggested that vimentin-expressing cells at the wound edge may be affected in the *stat3* mutants. To test this, we outcrossed *stat3* mutant with the *vimentin* reporter (*LeBert et al., 2018*) and examined the presence of vimGFP-positive cells at the wound following simple transection at 72 hptt. The number of vimGFP-positive cells were reduced in the *stat3* mutants compared to wild-type larvae (*Figure 4C,D*). The vimGFP-positive cells at the wound did not arise early and then disappear by 72 hptt, as their numbers were also lower at 24 hptt in the *stat3* mutants (*Figure 4—figure supplement 1C,D*). We did not evaluate the number of vimGFP-positive cells in stat3+/- larvae as we did not detect a defect in tissue regrowth in the heterozygotes (*Figure 4A,B*). These data suggest that Stat3 regulates the presence of vimentin-expressing cells specifically during wound responses, as vimGFP-expressing cells along the edge of the caudal fin of age-matched unwounded larvae appear unaffected in the *stat3* mutants (*Figure 4—figure supplement 1A*). Furthermore, the wound edge of the *stat3* mutants was smoother and appeared to lack collagen-based projections (*Figure 4A*). Therefore we examined collagen fiber integrity and organization by SHG microscopy. Interestingly, while collagen fibers failed to grow and exhibited less projections in the *stat3* mutants, they appeared to have normal organization (*Figure 4C*). We quantified projections at the wound edge by measuring the wound contour as previously (*LeBert et al., 2018*) and found a significant reduction in the *stat3* mutants (*Figure 4E*), suggesting a reduction in projections. This is consistent with our previous observations that vimentin was necessary for the formation of these collagen-based projections at the wound edge (*LeBert et al., 2018*).

In addition to Stat3, we examined the role of TGFβ in regulating wound-associated vimentin-expressing cells, as TGFβ is known to be critical in wound healing and modulate STAT3 signaling (*Aztekin et al., 2019*; *Derynck and Budi, 2019*; *Lichtman et al., 2016*; *Pakyari et al., 2013*). Using ALK5/TGFβ receptor I inhibitor, we found that disrupting the TGFβ pathway leads to impaired tissue regrowth and significantly diminished number of vimGFP-positive cells at the wound edge following tail transection (*Figure 4F–H*). This outcome appears to be wound-specific, as the tail fin area and vimGFP-positive cells in age-matched unwounded control appear normal (*Figure 4—figure supplement 1E,F*). Since TGFβ is known to modulate STAT3 signaling (*Derynck and Budi, 2019*), in future studies we plan to examine whether TGFβ regulates the dynamics of wound-associated vimentin-expressing cells via Stat3. It will be interesting to examine whether dysregulated Stat3 and/or TGFβ signaling contributes to the delay in wound-associated vimentin-expressing cells and impaired healing in the infected tail wound. Furthermore, it will be important to address the origin of these wound-associated vimentin-expressing cells, as that remains currently unknown. Our collective observations, that diminishing vimentin-expressing cells at the wound edge either by infection, or disrupting Stat3 or TGFβ pathways leads to impaired wound healing, support the idea that wound-associated vimentin-expressing cells are necessary for proper wound healing.

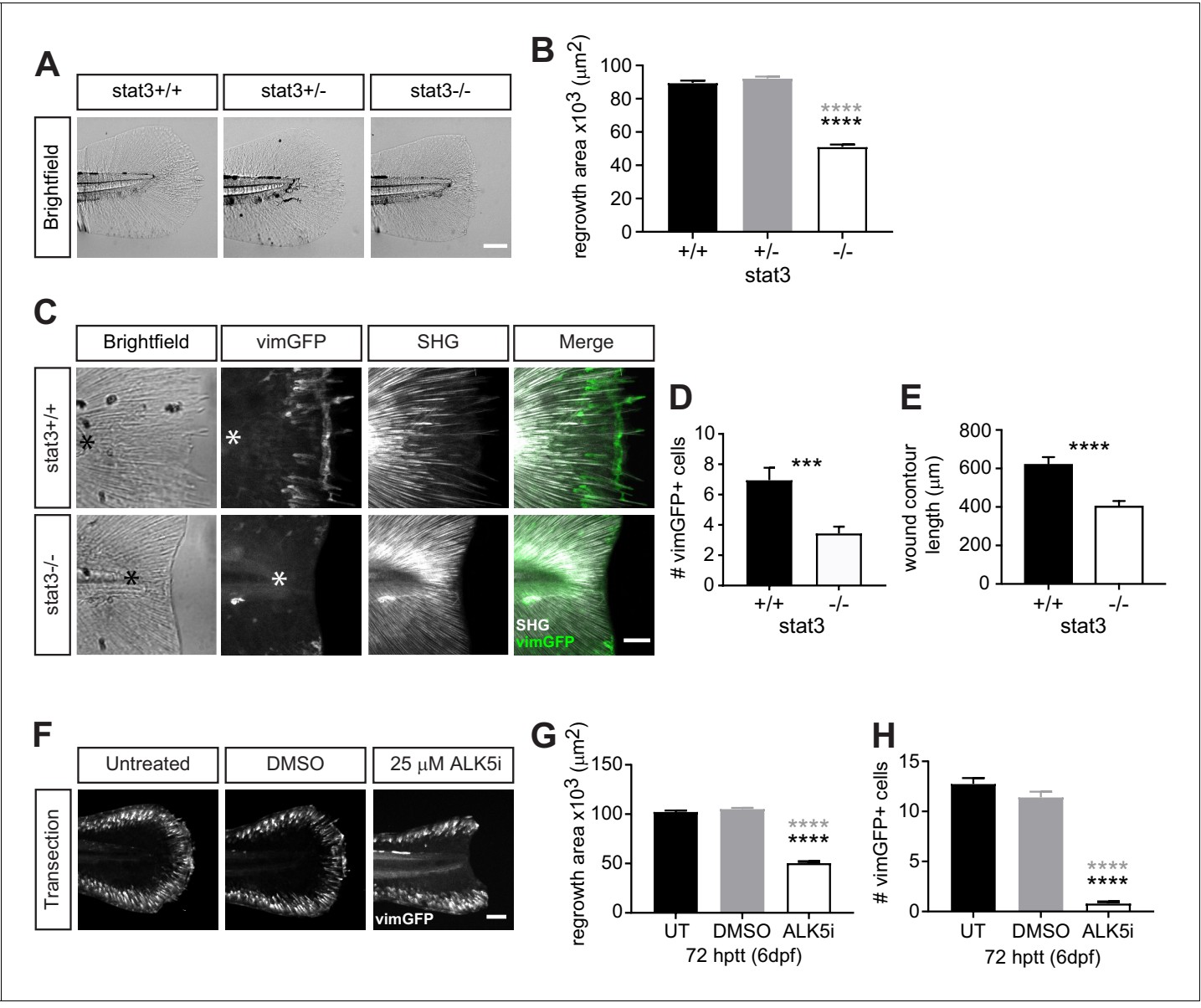

**Figure 4.** Wound healing and wound-associated vimentin-expressing cells are regulated by Stat3 and TGFβ signaling. (**A**) Single-plane brightfield images of wild-type, heterozygous and *stat3* mutant larvae showing tissue regrowth of caudal fins at 72 hr post tail transection (hptt) and (**B**) corresponding quantification of area of tissue regrowth. Values are arithmetic means and SE from three biological replicates. p values obtained by analyzing ranks due to residuals not being normally distributed. N = 41–50 larvae from het incross, per biological repeat, followed by genotyping. ****p<0.0001; black and gray * depict p values in comparison to wild-type and heterozygous, respectively. (**C**) Z-projections of multiphoton microscopy images showing the location of vimGFP-positive cells and SHG fiber organization in wild-type and *stat3* mutant larval caudal fins at 72 hptt. Asterisk denotes region just posterior to tip of notochord. Scale bar is 50 microns. (**D**) Quantification of the number of vimGFP-positive cells at the wound edge in wild-type and *stat3* mutant larvae. Values are least square means and SE from three replicates; Total N = 28–32 larvae per treatment. ***p<0.001 (**E**) Graph comparing the contour length of the wound edge, measured using SHG images, between wild-type and *stat3* mutant larvae. Values are arithmetic mean and SE from three experimental replicates with associated p values obtained by analyzing ranks due to residuals not being normally distributed. ****p<0.0001. Total N = 28–32. (**F**) Sum-projections of z-stacks acquired by spinning disk confocal microscope using *Tg(vim:GFP)* larvae, following tail transection at 72 hptt (6 dpf) in E3 medium only (untreated, UT) or in the presence of 25 µM ALK5 (TGFβRI) inhibitor or 0.5% DMSO as vehicle control. (**G**) Corresponding quantification of area of tissue regrowth. Values are arithmetic means and SE from three biological replicates. p values obtained by analyzing ranks due to residuals not being normally distributed. (**H**) corresponding quantification of vimGFP-positive cells at the wound edge. Values are least square means and SE from three biological replicates, with associated p values. Total N = 36–43 larvae for each treatment in (**G**) and (**H**). ****p<0.0001; black and gray * depict p values in comparison to untreated and DMSO-treated respectively.
DOI: https://doi.org/10.7554/eLife.45976.033

The following source data, source code and figure supplements are available for figure 4:

*Figure 4 continued on next page*

*Figure 4 continued*

**Source data 1.** Related to *Figure 4B*.
DOI: https://doi.org/10.7554/eLife.45976.036
**Source data 2.** Related to *Figure 4D*.
DOI: https://doi.org/10.7554/eLife.45976.037
**Source data 3.** Related to *Figure 4E*.
DOI: https://doi.org/10.7554/eLife.45976.038
**Source data 4.** Related to *Figure 4G*.
DOI: https://doi.org/10.7554/eLife.45976.039
**Source data 5.** Related to *Figure 4H*.
DOI: https://doi.org/10.7554/eLife.45976.040
**Source code 1.** Related to *Figure 4B,E and G*.
DOI: https://doi.org/10.7554/eLife.45976.041
**Source code 2.** Related to *Figure 4D and H*.
DOI: https://doi.org/10.7554/eLife.45976.042
**Figure supplement 1.** Stat3 or TGFβ signaling does not affect the presence of vimGFP-positive cells associated with development.
DOI: https://doi.org/10.7554/eLife.45976.034
**Figure supplement 2.** Table summarizing the characteristics of the three wound models.
DOI: https://doi.org/10.7554/eLife.45976.035

In summary, here we present the burn wound and the infected transection as alternative tail wound models in larval zebrafish with characteristic inflammatory and tissue responses that are different from the classical tail transection (summarized in *Figure 4—figure supplement 2*). The burn wound was associated with an intermediate inflammatory response, loss of collagen fibers with no effect on vimGFP-positive cells, and delayed healing as compared to tail transection. The infected wound had the least regenerative capacity with the most extensive inflammation and tissue damage. Most striking was the loss of vimGFP-positive cells and impaired collagen reorganization that correlated with the defect in repair. In addition, we identified Stat3 and TGFβ as important signaling pathways in regulating the dynamics of vimentin-expressing cells during wound response, since disrupting either pathway leads to diminished numbers of vimentin-expressing cells at the wound site accompanied by impaired healing. Collectively, the wound models presented here expand our toolkit to study tissue healing and regeneration, and provide a useful in vivo approach to investigate mechanisms of crosstalk between the innate immune system and the remodeling machinery in response to complex injuries.

## Materials and methods

**Key resources table**

| Reagent type (species) or resource | Designation | Source or reference | Identifiers | Additional information |
|---|---|---|---|---|
| Strain, strain background (10403S) | *Listeria monocytogenes* (strain 10403S) | *Vincent et al., 2016* | | was be obtained from JD Sauer Lab, University of Wisconsin - Madison |
| Strain, strain background (D. Rerio) | *WT (AB)* | ZIRC | ZL1 | https://zebrafish.org /home/guide.php |
| Strain, strain background (D. Rerio) | *Tg(tnf:GFP)* | *Marjoram et al., 2015* | | was obtain from M Bagnat Lab, Duke University |
| Strain, strain background (D. Rerio) | *Tg(vim:GFP)* | *LeBert et al., 2018* | | can be obtained from A Huttenlocher Lab, University of Wisconsin - Madison |
| Strain, strain background (D. Rerio) | *Tg(lysC: mCherry-histone2b)* | *Lam et al., 2014* | | can be obtained from A Huttenlocher Lab, University of Wisconsin - Madison |

*Continued on next page*

*Continued*

| Reagent type (species) or resource | Designation | Source or reference | Identifiers | Additional information |
|---|---|---|---|---|
| Strain, strain background (D. Rerio) | *Tg(mpeg1:mCherry-histone2b)* | *Vincent et al., 2016* | | can be obtained from A Huttenlocher Lab, University of Wisconsin - Madison |
| Strain, strain background (D. Rerio) | *Tg(mpeg1: GFP-histone2b)* | this paper | | can be obtained from A Huttenlocher Lab, University of Wisconsin - Madison |
| Strain, strain background (D. Rerio) | *Tg(lysC:BFP)* | *Rosowski et al., 2018* | | can be obtained from A Huttenlocher Lab, University of Wisconsin - Madison |
| Strain, strain background (D. Rerio) | *Tg(mpeg1:mCherry-CAAX)* | *Bojarczuk et al., 2016* | | was obtained from LI Zon Lab, Boston Children's Hospital, Dana Farber Cancer Institute |
| Strain, strain background (D. Rerio) | *Tg(lysC:BFP/mpeg1:mCherry-CAAX)* | *de Oliveira et al., 2019* | | can be obtained from A Huttenlocher Lab, University of Wisconsin - Madison |
| Strain, strain background (D. Rerio) | *casper* | *White et al., 2008* | ZL1714 | https://zebrafish.org/home/guide.php |
| Strain, strain background (D. Rerio) | *stat3^{stl27/+}* | *Liu et al., 2017b* | | was obtained from L Solnica-Krezel Lab, Washington University School of Medicine, St. Louis |
| Recombinant DNA reagent (D. Rerio) | tol2-mpeg1:GFP-histone2b | this paper | | can be obtained from A Huttenlocher Lab, University of Wisconsin - Madison |
| Chemical compound, drug | ALK5 inhibitor (ALK5i) | Selleckchem | item no. SB431542; RRID:SCR_003823 | 25 µM in DMSO; no pretreatment, start treatment upon wounding, refresh daily |
| Software, algorithm | GraphPad Prism | | RRID:SCR_002798 | https://www.graphpad.com/scientific-software/prism/ |
| Software, algorithm | SAS | | RRID:SCR_008567 | https://www.sas.com/en_us/home.html |
| Software, algorithm | Fiji, ImageJ | *Schindelin et al., 2012* | RRID:SCR_002285 | https://fiji.sc/ |
| Software, algorithm | CurveAlign | *Liu et al., 2017a* | | https://loci.wisc.edu/software/curvealign |
| Other | High temperature cautery pen, fine tip | Bovie | AA01 | http://www.boviemedical.com/hightempcauteries/ |

## Ethics

Animal care and use was approved by the Institutional Animal Care and Use Committee of University of Wisconsin (protocol M005405-02) and strictly followed guidelines set by the federal Health Research Extension Act and the Public Health Service Policy on the Humane Care and Use of Laboratory Animal, administered by the National Institute of Health Office of Laboratory Animal Welfare.

## Zebrafish maintenance, handling and transgenic lines utilized

All protocols using zebrafish in this study has been approved by the University of Wisconsin-Madison Research Animals Resource Center (protocol M005405-A02). Adult zebrafish were maintained on a 14 hr:10 hr light/dark schedule. Upon fertilization, embryos were transferred into E3 medium and maintained at 28.5°C. For wounding assays, 3 days post-fertilization (dpf) larvae were anesthetized in E3 medium containing 0.2 mg/mL Tricaine (ethyl 3-aminobenzoate; Sigma-Aldrich). To prevent pigment formation, some larvae were maintained in E3 medium containing 0.2 mM *N*-Phenylthiourea (PTU) (Sigma-Aldrich). Adult AB strain fish and transgenic zebrafish lines including *Tg(tnf:GFP)* (*Marjoram et al., 2015*), *Tg(vim:GFP)* (*LeBert et al., 2018*), *Tg(lysC:mCherry-histone2b)* (*Lam et al., 2014*), *Tg(mpeg1:mCherry-histone2b)* (*Vincent et al., 2016*), *Tg(lysC:BFP/mpeg1:Cherry-CAAX)*

(*Bojarczuk et al., 2016*; *de Oliveira et al., 2019*; *Rosowski et al., 2018*), and mutant lines including *casper* (*White et al., 2008*) and *stat3^{stl27/stl27}* (*Liu et al., 2017b*) were utilized in this study. *Tg(vim: GFP)* line was outcrossed to *stat3^{stl27/+}* to generate a *vimentin* reporter line in the *stat3* mutant background. The *stat3* mutant line is maintained as *stat3^{stl27/+}* as described previously (*Liu et al., 2017b*) and consequently, all wound healing experiments were performed blinded using a minimum of 36 larvae with subsequent genotyping. For multiphoton microscopy imaging, torsos were used to identify genotype, and *stat3^{stl27/stl27}* and *stat3^{+/+}* tails were subsequently imaged.

## Generation of *Tg(mpeg1:GFP-histone2b)* zebrafish

GFP-histone2b flanked by BamHI and XbaI was derived from version generated in *Vincent et al. (2016)* and subcloned into mini-Tol2 vector (*Urasaki et al., 2006*) linked to mpeg1 promoter (*Harvie et al., 2013*). 3 nL of mixture containing 12.5 ng/µL DNA plasmid with 17.5 ng/µL Tol2 transposase mRNA was injected into one-cell stage embryo (*Lam et al., 2014*). Larvae at 3 dpf were screened for GFP expression and grown up to generate stable lines.

## Caudal fin transection and thermal injury

Transection of the caudal fin was performed using surgical blade (Feather, no. 10) at the boundary of the notochord without injury to the notochord. To perform thermal injury, larvae were placed in 60 mm milk-treated dish with E3 containing Tricaine. A fine tip cautery pen (Bovie, Symmetry Surgical, Antioch, TN) was used to burn the caudal fin. The cautery wire was placed into the E3 medium, held to the posterior tip of the caudal fin, and turned on for 1–2 s until tail fin tissue curled up without injuring the notochord. Following injury, larvae were rinsed with E3 medium and placed in fresh milk-coated dishes with fresh E3 medium, and maintained at 28.5˚C until live imaging for wound healing assay or fixed at indicated times as described below.

## Bacterial culture, preparation and tail wound infection

Unlabeled or mCherry-expressing *Listeria monocytogenes* strain 10403S was used in this study (*Vincent et al., 2016*). *L. monocytogenes* were grown in brain–heart-infusion (BHI) medium (Becton, Dickinson and Company, Sparks, MD). For infection experiments, a streak plate from frozen stock was prepared and grown at 37˚C. The day before infection, a fresh colony was picked and grown statically in 1 mL BHI overnight at 30˚C to reach stationary phase. Next day, bacteria were sub-cultured for ~1.5–2.5 hr in fresh BHI (4:1, BHI:culture) to achieve growth to mid-logarithmic phase (OD600 ≈ 0.6–0.8). Bacteria were washed three times in sterile PBS and resuspended in 100 µL of PBS. For wounding, larvae were placed in 5 mL E3 medium containing Tricaine in 60 mm milk-treated dish. 100 µL bacterial suspension, or 100 µL PBS for control uninfected wounding, was added to the E3 medium and swirled gently to achieve even distribution of bacteria. Caudal fin transection of larvae in control or infected E3 medium was performed as described above. Larvae were immediately transferred to a horizontal orbital shaker and shaken for 30 min at 750 rpm. Control and infected larvae were then rinsed five times with 5 mL E3 medium without Tricaine to wash away bacteria, and maintained at 28.5˚C until live imaging for wound healing assay or fixed at indicated times as described below. Larvae were not treated with antibiotics at any point during the experiment.

## Drug treatment

To study the role of TGFβ signaling in wound healing, ALK5/TGFβRI inhibitor was used at 25 µM working concentration (SB431542, Selleckchem, Houston, TX), a dose that we identified that did not obviously impair development, or 0.5% DMSO as vehicle control. 3 dpf larvae were wounded as described above, rinsed to remove Tricaine, and placed in E3 medium containing inhibitor or DMSO. E3 medium containing inhibitor or DMSO was refreshed daily.

## Fixation

Larvae were fixed in 1.5% formaldehyde (Polysciences, Warrington, PA) in 0.1 M Pipes (Sigma-Aldrich), 1.0 mM MgSO_4 (Sigma-Aldrich) and 2 mM EGTA (Sigma-Aldrich) overnight at 4˚C. Next day samples were rinsed with PBS at least once, and stored in PBS until imaging.

## Wound healing assay

Wild-type larvae were wounded as above and individual larvae were maintained in a 24-well plate at 28.5°C. Each day, larvae were placed in 60 mm milk-treated dish with E3 medium containing Tricaine, held in position using an in-house custom-shaped copper wire, and a single-plane image was acquired using Zeiss Zoomscope (EMS3/SyCoP3; Zeiss, Oberkochen, Germany; Plan-NeoFluar Z objective; 112X magnification (0.7 µm resolution, 2.1 mm field of view, 9 µm depth of field) and Zen software (Zeiss). Larvae were rinsed, placed into fresh E3 medium, and returned to 28.5°C until subsequent imaging. Tissue regrowth area was measured over time.

## Dissemination of *L. monocytogenes* in tail wound infection

To evaluate extent of dissemination during tail would infection, mCherry-expressing version of *L. monocytogenes* was used to perform tail would infection on PTU-treated wild-type larvae. Larvae were fixed at indicated times post infection and imaged by Zeiss Zoomscope (EMS3/SyCoP3; Zeiss; Plan-NeoFluar Z objective; 63X magnification (0.8 µm resolution, 3.7 mm field of view, 12 µm depth of field). 200 µm sized z-stacks with 5-micron step size were acquired by $1 \times 4$ tile imaging of the whole embryos. Images were collected and stitched post acquisition using Zen software (Zeiss).

## Microscopy for leukocyte recruitment, TNFα-positive and vimentin-positive cells

To observe leukocyte recruitment and TNFα expression at caudal fin injuries, PTU-treated triple transgenic lines (*Tg(tnf:GFP x Tg(lysC:BFP/mpeg1:mCherry-CAAX)*) were imaged using laser scanning confocal microscope (FluoView FV1000; Olympus, Center Valley, PA) with an NA 0.75/20x objective and FV10-ASW software (Olympus). Fixed samples were placed in Ibidi chamber in 0.1% Tween-20-PBS solution and a 5-micron step series was acquired where each fluorescence channel was collected sequentially, using 250 micron size pinhole, 8.0 µs/pixel pixel dwell time, and $512 \times 512$ resolution. Alternatively, in tail wound infection experiments PTU-treated transgenic larvae with nuclear (histone2b) labeling (*Tg(lysC:mCherry-histone2b) x Tg(mpeg1:GFP-histone2b)*) were used to quantify leukocyte recruitment in fixed samples, where single-plane images were collected using Zeiss Zoomscope (EMS3/SyCoP3; Zeiss; Plan-NeoFluar Z objective; 112X magnification (0.7 µm resolution, 2.1 mm field of view, 9 µm depth of field) and Zen software (Zeiss). *Tg(vim:GFP)* lines, either in PTU-treated wild-type or *casper* mutant background, were used to observe vimentin:GFP (vimGFP)-positive population of cells. Fixed samples were placed in Ibidi chamber in 0.1% Tween-20-PBS solution and a 4-micron step series was collected using spinning disk confocal microscope (CSU-X, Yokogawa, Sugar Land, TX) with a confocal scanhead on a Zeiss Observer Z.1 inverted microscope, Plan-Apochromat NA 0.8/20x objective, a Photometrics Evolve EMCCD camera and Zen software (Zeiss).

## Survival assay

Wild-type larvae were used to perform tail wound infection as described above. Following wounding, individual larvae were place in 96-well plate in E3 medium and maintained at 28.5°C. Larvae were observed daily up to 7 days post infection (dpi) and scored dead upon absence of heart beat.

## Multiphoton microscopy of second harmonic generation (SHG) and vimentin-expressing cells

As previously described (*LeBert et al., 2016*; *LeBert et al., 2015*) fixed caudal fin samples from *casper* mutant or PTU-treated larvae prepared by removal from the body with a scalpel blade (Feather #15) then imaged in a 50 mm coverglass (#1.5) bottom dish (MatTek, Ashland MA) with the glass bottom depression covered with a second coverslip to minimize sample movement. Caudal fins were imaged using a custom-built multiphoton microscope (*Conklin et al., 2011*; *LeBert et al., 2016*) at the Laboratory for Optical and Computational Instrumentation using a 40X long working distance water immersion lens (1.2 NA, Nikon, Melville NY). Backwards SHG was collected with the multiphoton source laser (Chameleon UltraII, Coherent Inc, Santa Clara, CA) tuned to either 890 nm, with a 445/20 nm bandpass emission filter (Semrock, Rochester NY), or 740 nm with 370/10 nm bandpass emission filter (Semrock). The fluorescent signal from the vimentin:GFP expression was sequentially collected using a 520/35 nm bandpass emission filter (Semrock) and both signals were

detected using a H7422P-40 GaAsP Photomultiplier Tube (PMT) (Hamamatsu, Japan). Brightfield images were simultaneously collected using a separate photodiode-based transmission detector (Bio-Rad, Hercules CA). Imaging parameters remained constant across imaging days for a given replicate. Data were collected as z-stacks with optical sections two microns apart, at 512 × 512 resolution.

## Quantification

### Tissue regrowth area measurement

Tissue regrowth area during wound healing was measured using FIJI (*Schindelin et al., 2012*) by outlining the total fin tissue area distal to the notochord using the polygon tool.

### Leukocyte recruitment and TNFα expression

Leukocyte recruitment was quantified in the caudal fin tissue area distal to the caudal vein loop by counting individual cells or area thresholding of fluorescence intensity using FIJI. TNFα expression was quantified by scoring cells TNFα-negative in the absence of GFP signal, or TNFα-positive when GFP signal was detected within macrophages, or by area thresholding. In area thresholding, a single slice from the z-stack of brightfield channel was selected to outline the area of measurement in the caudal fin using the polygon tool; this region of interest (ROI) was copied onto the sum z-projection of the z-stack from the corresponding macrophage (mCherry) channel; fluorescence intensity was thresholded to measure macrophage area within the outlined caudal fin area; macrophage area was outlined using ROI manager and copied onto the sum z-projection of the z-stack from the corresponding TNFα (GFP) channel, fluorescence intensity was thresholded to measure TNFα area within the outlined macrophage area (*Figure 2—figure supplement 2E*).

### Area devoid of fibers

To measure the area between the wound tissue edge and the ends of the SHG detected fibers, sum z-projections of multiphoton SHG z-stacks, with the notochord vertically centered, were generated using FIJI. A free hand line was drawn along the ends of the fibers. This line was applied to the corresponding brightfield image and the area between this fiber-end boundary and the wound edge was drawn and measured. Because these measurements were collected on z-projections, the curling up of the tissue with thermal injury was not taken into account, thus the measurements on those samples may be an underestimation of the actual area devoid of fibers.

### Fiber angle relative to wound edge

The angle of SHG detected fibers relative to the wound edge were quantified as previously described (*LeBert et al., 2018*). Briefly, sum z-projections of SHG image stacks (generated using FIJI) of the larval caudal fin were analyzed using CurveAlign software (*Liu et al., 2017a*), which extracts and measures various aspects of fiber orientation, including orientation to a user designated boundary. The corresponding bright field image was used to mark the wound edge boundary and fibers within 50 microns of the boundary were included in the analysis. Detection sensitivity was set to maximize fiber detection but with an emphasis on minimizing false positives. The sensitivity setting was maintained for all samples within an experiment.

### Vimentin-expressing cell counts

To assess the presence of vimentin-expressing cells at the wound edge, using FIJI sum z-projections of z-stacks collected using either multiphoton microscopy or spinning disk confocal microscopy were used to count the number of vimGFP-positive cells present within a 50 × 100 micron rectangle, with the long axis bordered on the wound edge and centered vertically in line with the notochord (*Figure 4—figure supplement 1A*). If any part of the cell was present within the rectangle it was counted.

### Wound contour length

To assess the relative convolution of wound edge as an indicator of projections, the length of the wound edge was measured using the freehand line tool in FIJI to trace the contour of the wound edge on the z-projected SHG image, as previously described (*LeBert et al., 2018*).

## Statistical analyses

All experiments in the main text and figures consist of at least three biological replicates. We define biological replicates as the inclusion of three separate clutches from three separate days. All data were graphed using Prism (GraphPad Software, Inc, San Diego, CA) with statistical analyses performed using SAS/STAT 9.4 (SAS Institute Inc, Cary, NC). Analysis of variance using the SAS proc mixed procedure was used for area devoid of fibers, fiber angle, macrophage area data and wound contour length, accounting for the variation due to fixed effects (condition, time, condition by time interaction) and random effects (replicate, replicate by treatment). For tissue regrowth area data, where measurements were made on the same larva over time, the same model was used but included a repeated measures statement with an ar(1) error structure to account for autocorrelated errors. If the normality assumption of errors failed, then a non-parametric analysis was performed using the ranks. Where appropriate, this distinction is indicated in the figure legends. Immune cell count data were analyzed using SAS GLIMMIX (General Linear Mixed Model) procedure, using the same model structure as above, with negative binomial distribution. If values failed to converge using negative bionomial distribution, Poisson distribution was used and it is indicated in the figure legends. For vimGFP-positive cell counts, a similar GLIMMIX procedure was used, stratified by time due to nonlinearity. p values are displayed as *<0.05, **<0.01, ***<0.001 and ****<0.0001 in the graphs. Actual p values are reported in the source data files.

## Acknowledgements

We thank Pui-ying Lam for technical assistance, and Peter Crump and Dr. Jens C Eickhoff for valuable guidance in statistical analysis.

## Additional information

### Funding

| Funder | Grant reference number | Author |
| --- | --- | --- |
| National Institute of General Medical Sciences | GM1 18027 | Anna Huttenlocher |

The funders had no role in study design, data collection and interpretation, or the decision to submit the work for publication.

### Author contributions

Veronika Miskolci, Conceptualization, Data curation, Formal analysis, Investigation, Methodology, Writing—original draft; Jayne Squirrell, Conceptualization, Data curation, Formal analysis, Investigation, Methodology, Writing—original draft, Project administration; Julie Rindy, Data curation, Formal analysis, Investigation, Methodology, Project administration; William Vincent, Investigation, Methodology; John Demian Sauer, Resources, Methodology, Writing—review and editing; Angela Gibson, Conceptualization, Writing—review and editing; Kevin W Eliceiri, Conceptualization, Methodology, Writing—review and editing; Anna Huttenlocher, Conceptualization, Formal analysis, Funding acquisition, Methodology, Project administration, Writing—review and editing

### Author ORCIDs

Veronika Miskolci https://orcid.org/0000-0001-7900-4626
Jayne Squirrell http://orcid.org/0000-0003-0651-6027
John Demian Sauer http://orcid.org/0000-0001-9367-794X
Kevin W Eliceiri http://orcid.org/0000-0001-8678-670X
Anna Huttenlocher https://orcid.org/0000-0001-7940-6254

### Ethics

Animal experimentation: This study was performed in strict accordance with the recommendations in the Guide for the Care and Use of Laboratory Animals of the National Institutes of Health. All of

the animals were handled according to approved institutional animal care and use committee (IACUC) protocols of the University of Wisconsin (protocol M005405-02).

## Decision letter and Author response

Decision letter https://doi.org/10.7554/eLife.45976.046
Author response https://doi.org/10.7554/eLife.45976.047

## Additional files

### Supplementary files

• Transparent reporting form
DOI: https://doi.org/10.7554/eLife.45976.043

### Data availability

All data generated or analyzed during this study are included in the manuscript and supporting files.

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
