## [Decision Letter]

Thank you for submitting your article "Distinct inflammatory and wound healing responses to complex caudal fin injuries of larval zebrafish" for consideration by *eLife*. Your article has been reviewed by three peer reviewers, one of whom is a member of our Board of Reviewing Editors, and the evaluation has been overseen by Didier Stainier as the Senior Editor. The reviewers have opted to remain anonymous.

The reviewers have discussed the reviews with one another and the Reviewing Editor has drafted this decision to help you prepare a revised submission.

Summary:

The classical model of tissue repair in zebrafish larva is fin transection. The Huttenlocher group recently extended this model to more complicated cases of wounding burns (published previously). In the follow up study presented here they carefully compared different wound responses in three models: transections; burns; and infected transections. Readouts include area of tissue outgrowth, neutrophil and macrophage infiltration, TNFα expression, and collagen (vimentin-GFP) matrix organization. The kinetics and magnitude of these diverse wound responses are distinct among these healing paradigms, with infection having the largest negative impact on healing. The reviewers find that conclusions are supported by the data and that the detailed temporal and spatial description of basic events at these different wound types should be of interest to the tissue repair community. However, they have raised concerns about the descriptive nature of the study, lack of mechanistic insight, and low levels of neutrophil recruitment.

Essential revisions:

1) The paper would benefit from a more comprehensive description and more mechanistic insight, perhaps actual cellular level look at re-epithelialization and epidermal morphology near these wounds, and some tie of one or more of these processes to known molecular pathways that orchestrate healing. An example might be H_2_O_2_, explored by this group before. Can differences in H_2_O_2_ production account for any of the observed differences in inflammatory responses?

2) According to Figure 1E, the tissues did not growth between 72-168 hpw. Figure 1—figure supplement 1A and Figure 1D and shows that the bacteria levels were very low at 120 and 168 hpw, respectively. However, it may be helpful to examine the 96 hpw since this may also clarify the temporal relationships between the infection clearance and the reappearance of vimGFP-positive cells in Figure 4—figure supplement 1B, C.

3) Figure 2A: the level of neutrophil recruitment in the transection model is surprisingly low. Also, the claimed second wave of neutrophil recruitment at 72 hpw is not obvious. The authors should provide additional data or more compelling statistical analysis to highlight the differences in neutrophil (and macrophage) recruitment.

4) Subsection “Thermal injury and infection trigger distinct inflammatory responses”: it is claimed that the reduction of TNFα-positive macrophages coincides with recovering of tissue damage. However, there is no recovery according to Figure 1E.

5) Figure 4: the nature and origin of the vimentin-positive cells is very unclear. Why is their appearance delayed in the infection model? Are they recruited to the wound or were already present there? More information about the identity of these cells is required, for example, lineage tracing or single cell transcriptomics.

---

## [Author Response]

Essential revisions:1) The paper would benefit from a more comprehensive description and more mechanistic insight, perhaps actual cellular level look at re-epithelialization and epidermal morphology near these wounds, and some tie of one or more of these processes to known molecular pathways that orchestrate healing. An example might be H_2_O_2_, explored by this group before. Can differences in H_2_O_2_ production account for any of the observed differences in inflammatory responses?

We strongly agree with this comment and believe that these are important aspects to address. Our lab is actively pursuing the mechanistic aspects that contribute to the commonalities and differences observed in these wound models. Since we found that infection both impaired the appearance of vimentin-positive cells and regeneration, we focused on identifying pathways that mediate induction of vimentin-positive cells at the wound edge with tail transection.

In the revised manuscript we identified STAT3 and TGFβ as playing a role in the regulation of vimentin-positive cells at the wound environment. We found that disruption of both of these pathways leads to reduction of vimentin-positive cells and defect in tissue regrowth following tail transection, providing important mechanistic insight into these processes. We include this data in the subsection “Disrupting STAT3 or TGFβ pathways impairs wound healing and diminishes wound associated vimentin-positive cells”, and it is shown in the new Figure 4. To accommodate this additional data, the content of the original Figure 4 has been moved to Figure 3 and Figure 3—figure supplement 1. New experiments are reflected in the Abstract, Introduction and Materials and methods sections as well. We plan to study the role of these pathways more in depth, and how they may be altered in the burn and the infected wounds, in future studies. We also plan to address whether TGFβ signals via STAT3 to regulate the presence of vimentin-positive cells at the wound. However, this major undertaking is beyond the scope of this current short report. Also see response to major point #5.

2) According to Figure 1E, the tissues did not growth between 72-168 hpw. Figure 1—figure supplement 1A and Figure 1D and shows that the bacteria levels were very low at 120 and 168 hpw, respectively. However, it may be helpful to examine the 96 hpw since this may also clarify the temporal relationships between the infection clearance and the reappearance of vimGFP-positive cells in Figure 4—figure supplement 1B, C.

We have included this data. We now show the full time course for the infected wound in Figure 1C.

3) Figure 2A: the level of neutrophil recruitment in the transection model is surprisingly low. Also, the claimed second wave of neutrophil recruitment at 72 hpw is not obvious. The authors should provide additional data or more compelling statistical analysis to highlight the differences in neutrophil (and macrophage) recruitment.

We have removed the claim about the second wave of neutrophil recruitment. In terms of the level of neutrophil recruitment, the values shown in this current manuscript are consistent with what we have previously reported in several recent works from our lab (Powell et al., Cell Reports 2017; Barros-Becker et al., JCS 2017; Rosowski et al., JI 2016; Tauzin S et al., Cell Biol 2014). However, we are aware of the variability in wound responses in the zebrafish community, which may be due to several factors, including specific fish facility, type of wound and individual handling of zebrafish embryos. In addition, these low numbers may be attributed to the way the tail transection is performed. All wounds in the current manuscript are performed without injury to the notochord. We are also studying tail wounds that include injury to the notochord, which elicit a larger inflammatory response. We stated in the Materials and methods section that the wounds are performed without injury to the notochord, (subsection “Caudal fin transection and thermal injury”). To further emphasize this information, it is also stated now in the experimental schematic in Figure 1A. We consider tail transection with injury to the notochord to be different and more complex compared to tail transection without injury to the notochord.

4) Subsection “Thermal injury and infection trigger distinct inflammatory responses”: it is claimed that the reduction of TNFα-positive macrophages coincides with recovering of tissue damage. However, there is no recovery according to Figure 1E.

We have modified the text to clarify this point and apologize for any confusion. We made this claim in regards to the burn wound, not infected wound. We now tried to clarify this observation in the subsection “Thermal injury and infection trigger distinct inflammatory responses”.

5) Figure 4: the nature and origin of the vimentin-positive cells is very unclear. Why is their appearance delayed in the infection model? Are they recruited to the wound or were already present there? More information about the identity of these cells is required, for example, lineage tracing or single cell transcriptomics.

We strongly agree that these are extremely interesting questions and we are actively trying to identify the origin of vimentin-positive cells. Our previous work (LeBert et al., 2018) indicates that the vimentin cells do not migrate into the wound site but rather arise there. In addition, in terms of lineage tracing that work also showed that these cells did not appear to arise from epithelial cells or macrophages, but further work will need to be done to define their origin. While we are excited about pursuing the origin of these cells, we are unable to address this question within the scope of this current manuscript. Currently we are developing the tools to be able to carry out some of the suggested approaches, such as cell transcriptomics. We also agree with the reviewers that the delay in the vimentin-positive cells at the infected wound is intriguing. Multiple factors may contribute to that outcome, including cell death and excessive inflammation. We have included some initial data in Figure 3—figure supplement 2 using a mutant form of *Listeria* with attenuated virulence where we show that infection with the mutant *Listeria* induces much reduced inflammation and no delay in the vimentin-positive cells. This suggests that excessive inflammation is a contributing factor in the observed delay, further suggesting a potential crosstalk between immune cells and vimentin-positive cells, an area we plan to further explore. Based on having identified STAT3 and TGFβ to play a role in the appearance of vimentin-positive cells at the wound edge (see response to major point #1), we are currently in the process of examining the possibility that STAT3 and TGFβ signaling is disrupted at the infected tail wound. These new wound models provide the opportunity to address interesting questions in wound healing and we are excited to utilize these models to further explore these questions.